# PV Penetration under Market Environment and with System Constraints

**Aris Dimeas** [1,*] **and George Kiokes** [2]

1   Power System Laboratory, School of Electrical and Computer Engineering,
    National Technical University of Athens 9, Iroon Polytechniou St, 15780 Athens, Greece
2   Laboratory of Electrical Machines and Installations, Division of Electrical, Electronics and Informatics,
    School of Engineering, Merchant Marine Academy of Aspropyrgos, 19300 Aspropyrgos, Greece
*   Correspondence: adimeas@power.ece.ntua.gr

**Abstract:** The installed capacity of PVs in the distribution grid is affected not only by network constraints, but also by the economic viability of the related investments. Depending on the market participation models, this is determined critically by the Day Ahead Market (DAM) prices. Increasing RES installations in a country usually results in a long term drop in the market prices and, as a consequence, a reduction in the income of the PVs investors and possible market cannibalization. This paper models the effect of large-scale penetration of PVs on the market prices and identifies the optimal penetration level for the viability of PV projects. The optimal penetration is highly related to the installation of new PVs and this is a parameter for the analysis. Therefore, the paper identifies different penetration costs for the different installation cost. Furthermore, the PV network hosing capacity can be increased by distribution network reinforcements. Therefore, in the paper, the investments for enhancement of the distribution grid are assessed with respect to market prices and are analyzed at the macroscopic level. Again, the analysis considers different costs for network reinforcements.

**Keywords:** PV; day ahead market; aggregated bid curves; investment cost; market cannibalization; res penetration; network reinforcement





## 1. Introduction

The EU has set highly ambitious goals for RES penetration in the European energy system until 2030, as described in the Fit for 55 [1] and the REpowerEU programs [2]. To achieve the goals, the installed capacity of PVs in the distribution networks needs to be drastically increased. Most of European countries participate in the Coupled Electricity Market. The term "coupled" implies a common market, having as the main goal the unification and reduction in market prices across Europe. Market coupling systems (Price Coupling of Regions-PCR and Continuous Intraday Market-XBID) operate in both day-ahead trading and intraday markets. EUPHEMIA is the algorithm that has been developed to solve the problem associated with the coupling of the day-ahead power markets in the PCR [3].

The goal of this paper is to evaluate the impact of large-scale PV penetration in market prices and future PV investments by analyzing real bid ask price curves and identifying the optimal PV penetration. Furthermore, the paper considers that the majority of the PVs will be installed in the distribution grid, thus, the main constraint is the capacity of the radial network or the substation.

The modeling of the electricity prices is complex, and several researchers are working on this area. The effect of RES participation on market prices is described, among others, in [4,5]. Other approaches adopt multi agent models such as [6,7] without analyzing bid curves. In [8–10], empirical methods are used to analyze real data assuming that there are no subsidies by the state and not focusing on scenarios with large-scale RES

penetration. Furthermore, several papers focus on the electricity price forecasting based on data series analysis, as summarized in [11–13]. These papers use statistical methods, based on linear regression or machine learning algorithms, such as deep learning. More complex approaches combine several different algorithms for decompositions, feature selection, clustering, and optimization for model training. These methods are not suitable for the scope of this paper since they cannot provide accurate results for cases that are very different from the dataset used for training. Namely they cannot provide estimations for completely different systems or scenarios, e.g., a system with twice RES as much installed capacity.

Other approaches are focusing on the analysis of the Coupled Electricity Market such as [14], where a consecutive market and network simulation approach has been adopted. This approach allows investigating grid investment, re-dispatch, and renewable energy source (RES) curtailment requirements under various levels of RES penetration. Other similar approaches, focusing on the transmission grid constraints are presented in [15–17]. These approaches are different to the case in this paper. This paper focuses on the distribution network and different constraints are applicable.

There are also approaches focusing only on distribution grid constraints and namely the possible voltage problems [18–20]. More specifically the overvoltage problems are analyzed taking into consideration different standards or regulations in different countries. In similar approaches the existence of on load tap changers (OLTCs) is considered in [21,22]. However, these approaches focus only on a specific location and cannot model the impact on the electricity prices.

There are also macroscopic approaches such as in [23] using a well-known model called PRIMES [24]. PRIMES simulates the European system, having detailed models for every country. It is a very complicated system that is not used for investment planning but for the design of the long-term EU policy. Namely for strategic decisions until 2050 or longer. The approach presented here is less complicated and does not require the creation of a model for each EU country.

In this paper the PV penetration limits in respect with market prices will be analyzed. More specifically the paper analyses the limits of PV penetration to avoid market cannibalization [25]. Market cannibalization is the case in which the oversupply of RES energy pushes the electricity prices to extremely low levels. Generally, when cheap RES energy is injected in the system, the more expensive thermal power stations reduce their production. This forces the electricity prices to go to lower levels. However, assuming that PVs are compensated with market prices, this may cause problems to the viability of the investments. Finally, the paper analyses the possibility to invest on network reinforcements, to further increase the RES penetration.

This paper adopts a different approach than the literature: after the creation of the models of the price curves, it uses mixed integer linear programming to identify a realistic level of RES penetration to avoid cannibalization. Beyond this limit the investments will become non-viable due to low prices. The technical constraints considered in this problem are described in the next two sections.

## 2. Problem Formulation

We assume that new PV installations are remunerated from the day ahead market only and the PV owners bid in the market without long term contracts. Furthermore, the instant RES penetration cannot exceed the technical limit set by network constraints. The PV investors may accept some curtailment if the yearly income is satisfactory. The increase in the RES penetration, however, reduces the prices in the Day Ahead Market (DAM), and this results in a reduction in the income of the PV owners.

To define the satisfactory income for the investor, we introduce the term reference price. This is the fixed price for the whole year that ensures the viability of the investment assuming no curtailment. Obviously, this price reflects the recovery of the installation cost and is a parameter for our problem. In the following analysis a range of values for the refence price is assumed.

In order to increase the hosting capacity, the investor may choose to pay for network reinforcements. Again, the prices in EUR/MW are a parameter and are associated with the depreciated cost for network investments. The amount of capital spent for network reinforcement is assessed against the viability of the increased PV installation investment.

### 3. Mathematical Formulation

Objective function:

The problem is a maximization problem, and the objective function is described by the following equation:

$$maximise\ f = \sum_{i}^{n}(Prod_i * Price_i) - AddNCap * Costf2Upgrd\ \forall\ i \in T \tag{1}$$

The goal is to maximize the revenues from the new PV capacity considering that investments for network reinforcements may be needed. The $CostforUpgrade$ is not the actual cost per MW but the residual cost, considering the depreciation in the investment. This is the general formulation of the problem. In the first example in Section 6 the investment cost is not considered.

Constraints

A set of constraints bound this mathematical problem. The first one is the maximum RES penetration allowed by the system operator and is described by the following inequality:

$$Prod_i + RES_i \leq PenLim * Load_i\ \forall\ i \in T \tag{2}$$

The system operator defines the penetration limit for each hour. For simplicity we consider that it is the same for the whole time.

The second constraint is related to the production at time step i, which should be equal to the nominal production minus the curtailment

$$Prod_i = NewCap * NrPVprod_i - Curt_i\ \forall\ i \in T \tag{3}$$

The next set of constraints are related to the profitability of the new PVs investments:

$$\sum_{i}^{n}(Prod_i * \text{Price}_i) \geq \sum_{i}^{n}(RefP * NewCap * NrPVprod_i)\ \forall\ i \in T \tag{4}$$

$$ResidL_i = Load_i - Prod_i - RES_i\ \forall\ i \in T \tag{5}$$

$$Price_i = g(ResidL_i)\ \forall\ i \in T \tag{6}$$

$$Price_i = a + b * ResidL_i + c * ResidL_i^2 + d * ResidL_i^3 + e * ResidL_i^4\ \forall\ i \in T \tag{7}$$

Constraint (4) sets the low limit for the revenues of the new PVs. The reference price $RefP$ is the minimum compensation price for a new PV, that could lead to an attractive investment assuming no curtailment. Constraints (5)–(7) are analyzed in the next section.

Finally, there is a set of constraints related to the capacity of the distribution grid:

$$Prod_i + RES_i \leq NewNCap\ \forall\ i \in T \tag{8}$$

$$NewNCap = CurNCap + AddNCap\ \forall\ i \in T \tag{9}$$

The production at any timestep cannot exceed the installed capacity. The surplus energy should be curtailed. The second part of (1) aims to increase this limit by network reinforcements in a cost-effective way.

Modeling of the DAM prices

The most challenging part of this work is the modeling of the relation between the additional PV production and the DAM prices. The starting point is the cumulative bid curves of a system.

Figure 1 presents the bid curves for an hour in the Greek electricity market (blue line). This curve cannot be directly used in an optimization problem.

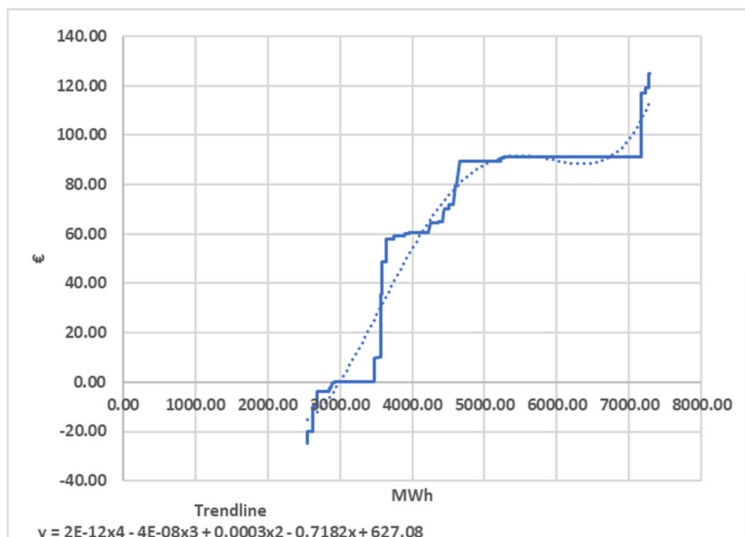

**Figure 1.** Bid Curves for the Greek Electricity market [5 May 2021 20:00–21:00].

The first step to model this curve was to perform a polynomial interpolation in the curve. The order 4 curve (dashed line in Figure 1) seems to fit quite well in the curve.

To insert this curve into the MILP problem, the constraint (6) was transformed to a piecewise linear constraint using the relevant functionality of the Gurobi optimizer [26]. This, however, has increased significantly the computation time. Running the model for a period of one year the problem was almost impossible to solve. Therefore, Equation (7) was also used. The results of (6) and (7) were comparable for a period of 1 month and the computation time using (7) was acceptable. The results of the next section are based on (7). The parameters a, b, c, d, and e are calculated using polynomial interpolation, as presented in Figure 1. The period and the origin of the processed data is mentioned in Section 6.

## 4. Impact of the Interconnections

Greece joined the pan-European electricity market at end of 2020. The market model supports the exchange of energy between countries (bidding zones) and the only constrain is the capacity of the interconnections. So, energy from cheap bidding zones flows to those with higher cost. Thus, it is important to analyze the effect of the interconnections in the market prices. For this work the approach is to estimate the flows between Greece, Italy and Bulgaria. The toolbox for machine learning toolbox of Microsoft Visual studio was used in this analysis. The toolbox ML.Net Model Builder provides the ability to test different algorithms and identify the most suitable one.

For the training of the modules, historical data for the flows between the countries, DAM prices, the system load and RES production were used. In Section 6 more details about the dataset are presented. Figure 2 presents the accuracy of the module for the flow from Greece to Italy, and Figure 3 the opposite flow. The main metric is the $R^2$, the coefficient of determination. For a perfect estimation, this metric is 1. From Figures 2 and 3, the estimation for the flow from Italy to Greece is more accurate. This is related to the amount of data since the flows from Italy to Greece are more frequent. Similar are the results for the interconnection with Bulgaria.

```
|                         Top 5 models explored                                              |
----------------------------------------------------------------------------------------------
|    Trainer              RSquared Absolute-loss Squared-loss RMS-loss Duration #Iteration    |
|54  LightGbmRegression   0.7325   84.89          12575.14     112.14   0.7      54           |
|16  FastTreeRegression   0.6448   98.23          16702.20     129.24   0.1      16           |
|76  FastTreeRegression   0.6375   93.87          17045.76     130.56   0.1      76           |
|73  FastTreeRegression   0.6343   98.57          17196.62     131.14   0.1      73           |
|85  LightGbmRegression   0.6302   102.08         17386.54     131.86   3.2      85           |
----------------------------------------------------------------------------------------------
```

**Figure 2.** Training results for the flow from Greece to Italy.

```
|                        Top 5 models explored                                          |
-----------------------------------------------------------------------------------------
|    Trainer                RSquared Absolute-loss Squared-loss RMS-loss Duration #Iteration |
|62  LightGbmRegression       0.8237      430.72     330612.34   574.99      0.3        62   |
|51  LightGbmRegression       0.8203      437.47     336958.26   580.48      0.5        51   |
|58  FastTreeRegression       0.8203      436.76     336973.17   580.49      0.6        58   |
|59  FastTreeRegression       0.8181      457.91     341012.07   583.96      0.5        59   |
|47  FastTreeRegression       0.8097      465.37     356815.87   597.34      0.1        47   |
-----------------------------------------------------------------------------------------
```

**Figure 3.** Training results for the flow from Italy to Greece.

The system tested various Trainers such as FastTree and LightGBM. In both cases the LightGBM regression [27] method achieved the higher $R^2$. The Trainer required less than a second to converge.

The developed models were not integrated in the optimization problem, but an iterative approach was adopted. After each iteration the system load was corrected according to the flows in the interconnections. The process stopped when there was no change in the flows.

## 5. Development of the Tool

The DIEM platform [28] developed by Smart RUE [29] was used for these studies. In a business interested in energy market information, such as energy trading, (RES) production, and energy market analysis, the typical workflow includes acquiring data from public and private sources to, eventually, feed the business decisions and respective support tools, as depicted in the simplified Figure 4, below.

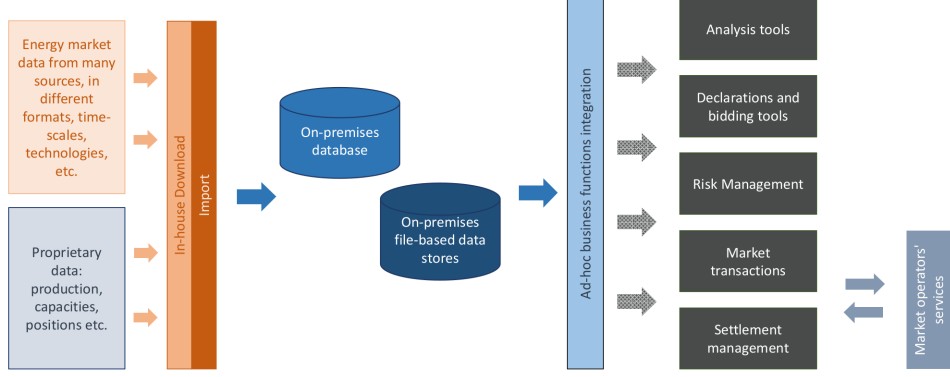

**Figure 4.** Typical Energy Trading Workflow.

Briefly, the data available through public sources (e.g., day-ahead market prices for one's own and neighboring markets, actual load and TSO (Transmission System Operator) official load forecasts, and generation per production type) are downloaded, aligned and stored, in a structured way, in a central repository (typically a relational DBMS), together with data acquired through private sources (such as own demand/production actuals and forecasts, proprietary market production forecasts, etc.) in order to form the basis of further analysis and processing by business specific tools and processes. Such tools are a BI (business intelligence) tool to analyze and produce insights, ETRM (energy trading and risk management) systems, and bidding optimization and market participation tools, which aid the automation and optimization of submitting the actual bid/ask orders to the respective market tool or optimally plan and execute the required hedging, according to the business strategy.

In today's highly correlated markets, it is of crucial importance to form an, as accurate as possible, view of the state of one's own as well as neighboring markets, in terms of demand, resources availability (renewable resources prediction and non-renewable resources costs and availabilities) and expected participation (which units are expected to offer their generation and at what marginal cost), so that the business can better define their strategy of participation in the different markets and hedging of future needs. This

need has been correctly identified by the EU and, acting upon it, ENTSO-E, which is the association of European TSOs, has defined a set of mandatory reports for TSOs and specific market participants to be shared with everyone interested, via its transparency platform. In its platform, data are organized, labeled, and aligned in several dimensions (e.g., areas, time resolution, and production types), so that they are easily combined together to form valued insights.

Since the ENTSO-E platform cannot possibly be gathering all data available via public sources, or, even, may be publishing late the official reports late (i.e., no earlier than the respective entity publishes them to the platform—still later with respect to when the information was first published to the original public source, which, in turn, may not be the TSO itself), it is of crucial importance for a business to have a dependable way to routinely acquire such information on a daily basis on-time. For information shared in an unstructured way (e.g., custom formatted excel workbooks, html tables, csv/txt files, different APIs, etc.), it is also important to label and align the data, using the same semantics so that everything can be combined together.

DIEM aspires to implement this first step of gathering market relevant data, on-time, and align and store them in its DBMS, offering them for immediate consumption by the interested parties, via a well-defined API (to directly feed third party systems/applications) and via an in-house developed web-app for simple visualizations and analysis by anyone. Figure 5, below, showcases the respective functionality that DIEM offers as basic infrastructure.

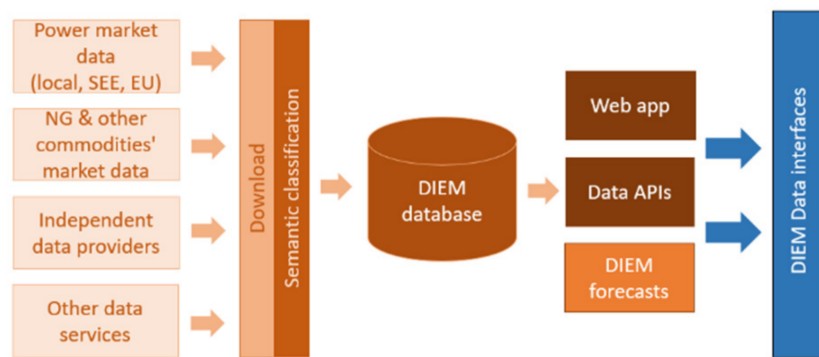

**Figure 5.** DIEM basic offering.

Additionally, the DIEM team is implementing and actively maintaining a set of forecasts for the Greek market, initially, based on state-of-the-art AI/ML (machine learning) and offering them via the platform (API and web app), so that they may be used in conjunction with the rest of the information. RES production forecasting requires weather forecasts, which DIEM acquires from different sources and with different granularities (including GFS and local weather forecasting services such as established university teams). The available forecasts are per bidding zone or control area for load, RES production and demand for intra-day, day-ahead (in fact, 48 h), week-ahead and month-ahead.

The platform also integrates private forecasts (i.e., own demand/production forecasts, etc.), visible and available only to the respective parties that have requested them; for doing so, DIEM may interface with a set of external systems, such as the SCADA or EMS of the respective parties. Custom forecasts are integrated in the platform after discussing and defining the project with the interested party for each managed RES portfolio (wind and solar), acquiring relevant system details (production unit details, historical production information, etc.) and agreeing on the period and methodology.

As already stated, DIEM interfaces with different systems, following best practices, to acquire and share data as presented in Figure 6:

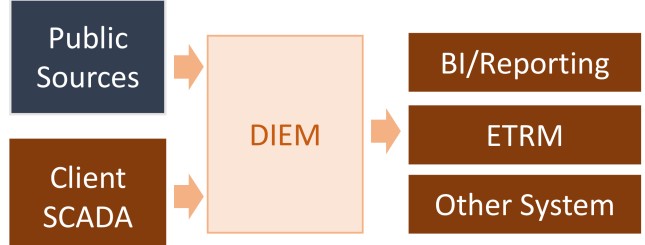

**Figure 6.** DIEM integrates with external systems for acquiring and sharing data.

API facilitates the exporting of data to client systems
All chart data in the web-app can be exported
Acquire managed RES data from client SCADA systems
Acquire client data from external data providers
The web site/app, offers the possibility to monitor selected datasets organized in custom dashboards (sets of graphs, defined by the user, which depict the selected time-series, updated automatically when the data in the DIEM database are updated).

## 6. Results

This section provides results using data from the Greek power system and the Greek electricity market. Greece is interconnected with all the neighboring countries, however only Italy and Bulgaria also participate also in the pan-European electricity market. Thus, the exchanges in these interconnections directly affect the prices in the day ahead market. Furthermore, Greece has a peak demand around 9500 MW while the installed capacity of wind farms is 3591 MW and PVs is 2398 MW. Regarding the PVs, they are mainly installed in the distribution network and several substations are congested. Therefore, this paper focuses on PV.

The data were collected from the Greek nominated market operator [30] and from ENTSOe [31]. Two time periods were selected:1 November 2020–31 October 2021 and 1 July 2021–30 June 2022. It should be noted that Greece entered the Pan-European coupled electricity market on 1 November 2020. Furthermore, the two periods were selected, in order to validate the model with high and low prices. Figure 7 presents the DAM prices for the selected periods. The second period is highly affected by the high natural gas prices.

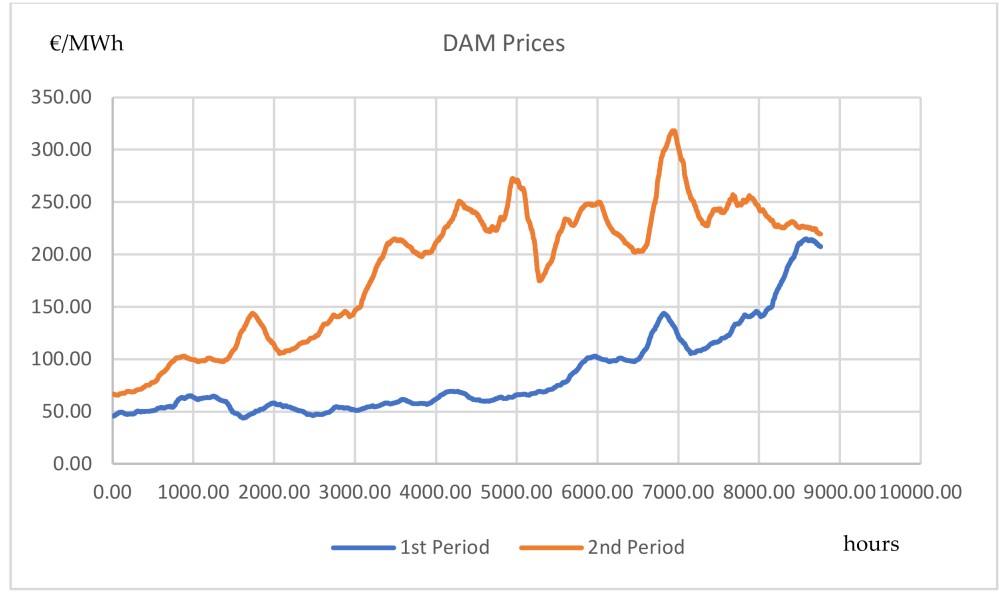

**Figure 7.** DAM prices during the 2 periods (€/MWh).

The data used for this work are:

- Aggregated bid curves (EnEx)
- Day ahead market prices
- Hourly load curve
- Hourly wind production
- Hourly PV production

The installed capacity of PVs is 2400 MW for the first period and 3.000 MW for the second. We assume that the network is congested. Thus, if the instant PV penetration exceeds 3000 MW then the surplus energy should be curtailed.

Scenario without network reinforcement

The first scenario analyses only the impact of the reference price in the curtailment. Mentioned before is the price the PV owner accepts to sell his production without curtailment. Receiving a price from the DAM, means that the owner would accept as long the total yearly income is higher or equal to the income with the reference price (and no curtailment). The results from this scenario are summarized in the following Tables 1 and 2:

**Table 1.** Results for the first period.

| Reference Price (EUR/MWh) | New PV Capacity (MW) | PV Production (MWh) | PV Curtailment (MWh) | Curt. (%) |
|---|---|---|---|---|
| 35 | 861.7 | 1,153,872.8 | 207,770.7 | 15% |
| 40 | 828.5 | 1,142,049.6 | 165,764.9 | 13% |
| 45 | 801.4 | 1,131,242.6 | 132,608.2 | 10% |
| 50 | 779.1 | 1,121,237.7 | 106,427.8 | 9% |
| 55 | 760.6 | 1,111,896.6 | 85,846.4 | 7% |
| 60 | 745.3 | 1,102,928.4 | 69,873.1 | 6% |
| 65 | 732.2 | 1,094,202.6 | 57,369.8 | 5% |
| 70 | 720.7 | 1,085,511.4 | 47,434.5 | 4% |
| 75 | 710.6 | 1,076,843.8 | 39,670.1 | 4% |
| 80 | 701.4 | 1,068,054.5 | 33,568.5 | 3% |

**Table 2.** Results for the second period.

| Reference Price (€/MWh) | New PV Capacity (MW) | PV Production (MWh) | PV Curtailment (MWh) | Curt. (%) |
|---|---|---|---|---|
| 35 | 1046.4 | 1,229,621.9 | 259,153.5 | 17% |
| 40 | 1008.4 | 1,220,496.6 | 214,113.9 | 15% |
| 45 | 978.1 | 1,212,766.7 | 178,710.2 | 13% |
| 50 | 953.0 | 1,206,051.2 | 149,740.1 | 11% |
| 55 | 931.8 | 1,200,148.7 | 125,487.2 | 9% |
| 60 | 913.9 | 1,194,976.4 | 105,267.9 | 8% |
| 65 | 899.0 | 1,190,481.9 | 88,566.6 | 7% |
| 70 | 886.4 | 1,186,404.0 | 74,738.4 | 6% |
| 75 | 875.5 | 1,182,580.6 | 62,948.7 | 5% |
| 80 | 865.9 | 1,178,939.6 | 52,952.6 | 4% |

There are two interesting observations from the two tables. The first observation is that a difference in the reference price does not affect proportionally the PV capacity. For example, the market prices in the second period were more than 2 times higher, however the new PV capacity is not significantly higher. This is related with the bid curve and the fact that after a certain installed capacity the prices become very low and the total income for the PV owners reaches close to zero.

The second observation is that if the reference price is low, the curtailment is quite high. In this case the increased curtailment is compensated by the higher PV capacity and the corresponding production.

In Figure 8 are the estimated prices for the case where the reference price is 80 EUR/MWh. The prices during the whole period are close to the reference price. If we had higher PV penetration, the prices would go beyond this level making the investments non-viable.

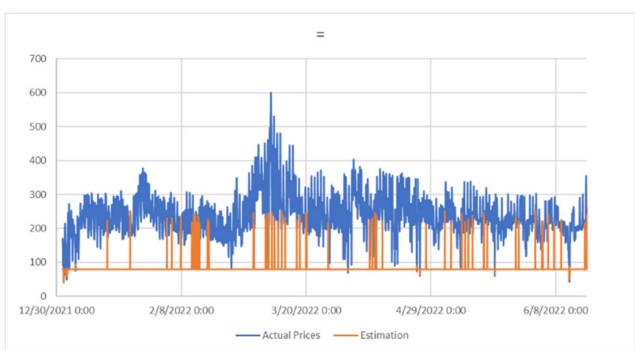

**Figure 8.** Estimated Prices for the second period when the reference price is 80 EUR/MWh (EUR/MWh).

Scenario with network reinforcement

The second scenario considers the investment costs for network reinforcements. The investment cost is the depreciated value for a twenty-year period. The starting point is based on the capital cost of PV installation, that is nowadays estimated at 600 thousand EUR/MW. Divided by 20 the result is EUR 30 thousand. Next, we consider different values around the previous figure. The Reference price for this scenario is assumed equal to 60 EUR/MWh.

The results in this case were similar as in the previous scenario and are presented in Tables 3 and 4. No big difference in the amount of the PV capacity was observed. In the case of low investment costs, the drop in the prices (due to the increased PV capacity were compensated by the reduction of the curtailment. When the investment cost is really high (e.g., 45.000 EUR/MW) the results are similar to the scenario 1.

**Table 3.** Results for the first period.

| Investment (EUR/MW) | New Network Capacity (MW) | New PV Capacity (MW) | Curt. (%) |
|---|---|---|---|
| 10,000 | 321.3 | 1066.3 | 2% |
| 15,000 | 293.2 | 1020.7 | 3% |
| 20,000 | 270.6 | 1010.4 | 4% |
| 25,000 | 255.4 | 983.6 | 4% |
| 30,000 | 210.9 | 942.7 | 5% |
| 35,000 | 145.2 | 882.5 | 5% |
| 40,000 | 92.3 | 827.8 | 6% |
| 45,000 | 21.1 | 764.2 | 6% |

**Table 4.** Results for the second period.

| Investment (€/MW) | New Network Capacity (MW) | New PV Capacity (MW) | Curt. (%) |
|---|---|---|---|
| 10,000 | 359.5 | 1236.7 | 4% |
| 15,000 | 309.3 | 1176.2 | 5% |
| 20,000 | 284.8 | 1148.9 | 5% |
| 25,000 | 276.0 | 1155.4 | 6% |
| 30,000 | 243.7 | 1108.2 | 6% |
| 35,000 | 149.6 | 1025.1 | 7% |
| 40,000 | 119.1 | 989.8 | 8% |
| 45,000 | 49.7 | 919.2 | 8% |

## 7. Conclusions

This paper analyses the correlation between PV penetration and market prices. The goal is to identify the limits of cannibalization. Additional capacity beyond this limit, does not only jeopardize new investments but also the existing ones under the same funding scheme. The key conclusion is that the significant increase in RES penetration could push market prices to lower levels. The result is that the PV owners will have reduced profit and their investments may not be viable after this penetration limit. The investors without PPAs and long-term contracts could put their investment at risk if they rely on spot market prices only. Furthermore, a significant parameter is the installation cost of PVs, which affects the total installed capacity. If the installation cost is low, additional PVs can be installed and the owners can afford extra RES curtailments. This parameter is critical for the investors, since if they have good projection of future PV installation costs, they can estimate how close the system is to the limit of cannibalization. Finally, the PV penetration could be further increased if the investors also cover the distribution networks reinforcement. If the reinforcement cost is relatively low the curtailment can be decreased. Again, if the investor can estimate future nominal costs for reinforcements, they can also estimate how safe their investments are.

**Author Contributions:** Conceptualization, A.D. and G.K.; Methodology, A.D.; Software, A.D.; Formal analysis, A.D.; Investigation, A.D. All authors have read and agreed to the published version of the manuscript.

**Funding:** This research received no external funding.

**Data Availability Statement:** All data have been retrieved from the DIEM platform. Online: https://diem-platform.com/. The platform allows one-week free access.

**Conflicts of Interest:** The authors declare no conflict of interest.

## Nomenclature

| | |
|---|---|
| $ResidL_i$ | Residual load at timestep i$i \in T = \{1, 2, \ldots N_t\}$ (MW) |
| $Load_i$ | System load at time I (MW) |
| $RES_i$ | Production of existing RES at timestep $i$ (MW) |
| $Prod_i$ | Production of the additional PVs at timestep $i$ (MW) |
| $Price_i$ | new DAM price formulated at time $i$ (€/MWh) |
| $RefP$ | Reference Price for PV installation to have acceptable IRR when the is no curtailment (€/MWh) |
| $Curt_i$ | Curtailment of the additional PV production at timestep $i$ (MW) |
| $NewCap$ | the additional PV capacity (MW) |
| $PenLim$ | Maximum Instant RES penetration (%) |
| $NewNCap$ | New total Network Capacity (MW) |
| $CurNCap$ | Current Network Capacity for RES (MW) |
| $AddNCap$ | Additional Network Capacity (MW) |
| $Cost2Upgd$ | The cost to increase Network Capacity (€/MWh) |
| $NrPVprod_i$ | the value of the normalized PV curve at timestep $i$ (0–1) |

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
