# Peer review of "PV Penetration under Market Environment and with System Constraints"

_energies, doi:10.3390/en15228673_

Round 1

Reviewer 1 Report

In this manuscript, the authors model the effect of large scale penetration of PVs on the market prices and identifies the optimal penetration level for the investors. However, following points need to be addressed.

1. The literature review section of the paper is very weak. The authors should include some recent research articles covering the topic addressed.

2. The authors should clearly present the research gap considering the recent research.

3. Please clearly mention the contributions of the proposed work.

4. The authors should improve the English language used in the manuscript.

5. Please properly write the Nomenclature section.

6. Figures included in the manuscript should be self-explanatory. Please mention the variables/quantities on X-axis and Y-axis of Figure 1.

7. Please insert clear and high quality figures in the manuscript.

8. Please define TSO?

9. Case study considered by the authors does not provide enough information of the system. Please provide details of the used system for better understanding of the reader.

10. The authors need to critically analyze and discuss the results of the case study.

11. Abstract and conclusion sections should be modified by including the key results and analysis of the designed methodology.

Reviewer 2 Report

Authors should address the following comments: 

1. Even though the paper is well-written, a revision from a native speaker is suggested to ensure that any grammatical errors are avoided. Particular attention has to be paid to the confusion of the words “its” and “each” and to the content of lines 209-212, which is a bit confusing.

2. It is highly recommended to change the format of the nomenclature (e.g. by formulating the data into a table) in order to enhance its comprehensiveness. Moreover, units of the presented variable should also be added to help the better understanding of the equations (for instance is “PenLim” expressed as a percentage, i.e. is it a dimensionless number?)

3. The literature review is quite limited. Some additions in this section are recommended in order to present a more thorough view of other relevant studies.

4. The novelty of this work is not clear. It is recommended to conclude the literature review by highlighting the new aspects this works covers in relation to past works.

5. Please check equation 3 in line 108, as it includes the variable “NormPVprod” which is not included in the Nomenclature.

6. The authors state that in order to reduce the computational time, equation 7 was used instead of equation 6. However, information either about the values of constants “a”, “b” and “c” of equation 7, or about the methodology followed in order to obtain these values is missing.

7. Details about the data used to obtain the Bid Curves for the Greek Electricity market of Figure 1 (for instance the reference years of these curves) are missing.

8. Although there is a reference to interconnections and energy flows between Greece and Bulgaria, there is no indication about the values of R2 metric of the corresponding Machine Learning Training. Are they comparable to those included in Figures 2 and 3?

9. Could the authors provide some details about the Machine Learning Training procedure, such as the Trainers tested, the Trainer which was finally selected, the duration and the reference years of the historical data used for the training etc.?

10. Please correct the cross-reference in page 6 / lines 250-251.

11. Please add axis titles and units in Figure 3 / page 7 / line 275. Please also correct the numbering of the respective figure as “Figure 3” has already been assigned to another Figure in page 5 / line 170.

12. Could the authors modify the variable names in Tables 1, 2, 3 and 4 so as to better reflect the corresponding variables? For instance, they could consider the use of “New PVs Capacity” instead of “Capacity”, “New PVs Production” instead of “Production”, “New Network Capacity” instead of “Network Capacity” etc. Alternatively, they could use the variable abbreviations as defined in the Nomenclature.

13. It is highly recommended to check the numbering of paragraphs throughout the text. For instance both “Introduction” and “Problem Formulation” sections share the number 1, while number 5 is assigned to “Development of the tool”, “Results”, “Next steps” and “Conclusions” sections. Particularly in “Results” section, the corresponding sub-sections are numbered as 4.1 and 4.2. Moreover, in line 96 there is a reference to Section 4, which seems no to be valid.

14. It is highly recommended to check the placement of references in the text, as some of them seem to be wrongly placed (for instance reference in line 144), while others are not mentioned at all (for instance references 6 and 7).

15. Although, the authors mention in the “Conclusions” section that the “increase of RES penetration could push market prices to lower levels”, no data about the market prices are presented in Tables 1-4. It is recommended to add such information in order to supplement the aforementioned statement. Furthermore, commenting about the variation of the curtailment with the new PV capacity and the associated energy production in the examined scenarios could be added in “Results” and / or in “Conclusions” section in order to further populate the results’ commenting.

Round 2

Reviewer 1 Report

The authors have improved the manuscript. However, the comments mentioned in the last review are not properly addressed and highlighted in the manuscript. For example:

1. The literature review section is still very weak. There are many research articles available in the literature regarding the topic of this paper.

2. In line 50-60, the authors mentioned the objectives of this paper. However, the comment was related to the research gap addressed in this paper considering the existing literature. Therefore, the authors should update the literature review section and considering that must present the research gap.

3. The main contribution of the paper still needs to be mentioned clearly. In line 56-60, the authors discussed other papers, not the work presented in this paper.

4. English language should be improved.

5. Information of the system considered in case study is not mentioned in lines 260-272.

6. Critical analysis and discussion of the results still missing.  

Author Response

Please check attached fie

Reviewer 2 Report

No further comments for the authors.

Round 3

Reviewer 1 Report

The authors have satisfactorily addressed all the comments.